

# Rapid detection of phase transitions
# from Monte Carlo samples before equilibrium

**Jiewei Ding[1,2], Ho-Kin Tang[3⋆] and Wing Chi Yu[1,2†]**

**1** Department of Physics, City University of Hong Kong, Kowloon, Hong Kong
**2** City University of Hong Kong Shenzhen Research Institute, Shenzhen 518057, China
**3** School of Science, Harbin Institute of Technology, Shenzhen, 518055, China

⋆ denghaojian@hit.edu.cn , † wingcyu@cityu.edu.hk

## Abstract

We found that Bidirectional LSTM and Transformer can classify different phases of condensed matter models and determine the phase transition points by learning features in the Monte Carlo raw data before equilibrium. Our method can significantly reduce the time and computational resources required for probing phase transitions as compared to the conventional Monte Carlo simulation. We also provide evidence that the method is robust and the performance of the deep learning model is insensitive to the type of input data (we tested spin configurations of classical models and green functions of a quantum model), and it also performs well in detecting Kosterlitz-Thouless phase transitions.



# 1  Introduction

The study of phase transitions in many-body systems is one of the hottest research topics in condensed matter physics. Microscopic constituents can couple and interact with each other in many different ways, giving rise to various phases of matter having intriguing macroscopic physical properties. Studying the transitions between different phases can give us deeper understandings of condensed matter physics especially in some non-trivial phases like topological phases where the order parameter is not readily available [1]. Monte Carlo(MC) simulation has been one of the most popular numerical techniques adopted to explore the physical properties of different phases in condensed matter models by the established Markov chain process. In addition, the large amount of data generated by MC simulations can be used for data-driven physics research, such as using machine learning to discover new physics from the data.

In the past few years, the classification of phases of matter using machine learning emerged as a prosperous research field in physics [2,3]. Recent studies have shown that data-driven machine learning models can classify different phases by finding unknown features in condensed matter models, and further locate the phase transition points using both supervised and unsupervised learning techniques. Unsupervised learning does not require prior labelling of the data. This is particular suitable for determining the number of phases in the phase diagram of new models. Examples of common unsupervised learning techniques include principal component analysis [4–6], meta-heuristic optimization [7], machine learning clustering [8] and deep autoencoder [6, 8, 9]. On the other hand, although supervised learning requires prior knowledge to label the training data, it can locate the transition points with high accuracy. Previous work has demonstrated the success of employing supervised learning in determining the phase transition points of, for example, the Ising models [10], the XY model [11] and the Hubbard model [12].

However, when using MC simulation, in some cases, it requires extensive computational resource to generate converged set of equilibrium data. For example, near the phase transition where critical slowing down occurs and thermal fluctuation diverges, or when we do quantum MC simulation, a longer time is also required for the simulation to reach equilibrium due to the computational complexity of the algorithm, which sometimes accompanied with the thorny sign problem [13]. To obtain the result in the thermodynamic limit, large system sizes are needed to locate the phase transition point accurately [14]. However, when the system size increases, the time required for simulation to reach equilibrium will also increase sharply, which results in a great increase in the time cost and computing resources for generating the training data. Our method being discussed here provides a novel approach to locate the phase transition points using the input MC data before equilibrium, thus saving the lengthy computation.

In this article, we tested some of the most up-to-date deep learning models, namely the Bidirectional Long Short-Term Memory (Bi-LSTM) and Transformer, which focus on analysing the time-domain data. The fact that deep learning has become a hot research area in re-

cent years has a lot to do with its success in image classification. AlexNet proposed by Alex Krizhevsky *et. al.* won the championship in an image classification competition in 2012, and its performance far outperformed other non-deep learning algorithms [15]. Later, it was found that deep learning algorithms are also outstanding in many tasks such as image generation, image segmentation and object detection. At the same time, people began to explore the application of deep learning on tasks involving time-series data, such as text translation and speech recognition and have proposed models such as Recurrent Neural Network (RNN) [16] and LSTM [17]. Transformer has shown great potential in the field of natural language processing in recent years, and many improved deep learning models based on Transformer prove it to be more suitable for extracting features from very long sequences than LSTM [18].

Unlike previous approaches which used a large amount of spin configurations generated from MC simulations after equilibrium is reached as the training data, here we used spin configurations from the first a few tens of MC steps as the sequential data input to the deep learning models. As a benchmark, we first employed our scheme to locate the phase transition in the two-dimensional (2D) Ising model on a square lattice. We then further extend the scope of data source and the condensed matter model with non-trivial phases like the topological phase in the XY model for the deep learning analysis. Surprisingly, we find our method not only works using spin configurations from classical MC, but also the Green functions obtained from quantum MC. In addition, we also compared the performance of Bi-LSTM and Transformer with other commonly used deep learning models, namely the Fully Connected Neural Network (FCN) and Convolutional Neural Network (CNN). We found that Bi-LSTM and Transformer performed far better than FCN and CNN in classifying the phases with MC data before equilibrium. We also find that the Bi-LSTM and Transformer can correctly classify the phases with smaller number of MC steps as compared to FCN and CNN.

The paper is organised as follows. In Section 2, we introduce our proposed method and the deep learning models in detail. We applied our scheme to detect the phase transition in the Ising model and compare the performance using different deep learning models in Sec. 3. Section 4 presents the results when employing our method to other condensed matter models with non-trivial type of phase transitions and to the quantum Hubbard model where inputs with the Green functions generated from quantum MC is used to determine the phase transition point. A conclusion is given in Sec. 5.

## 2 Deep learning models for time domain analysis

Figure 1 shows a schematic illustration of our proposed learning model. We use MC method to simulate the sequential data before reaching the equilibrium state and when the system is far away from the phase transition point as the training input for the deep learning models. Taking a classical spin model as an example, the sequence is formed by randomly selecting a site in the system and take its spin configuration in the first $m$ steps in the MC simulation. The deep learning model extracts features from the sequential data through the LSTM block or the Transformer block and performs binary classification of the phases. The trained deep learning model is then fed with data from the full driving parameter range to predict which phase the data belongs to. The phase transition point of the system is estimated from the deep learning model's output at the probability value of 0.5.

When one uses equilibrium spin configurations as the training data for the deep learning

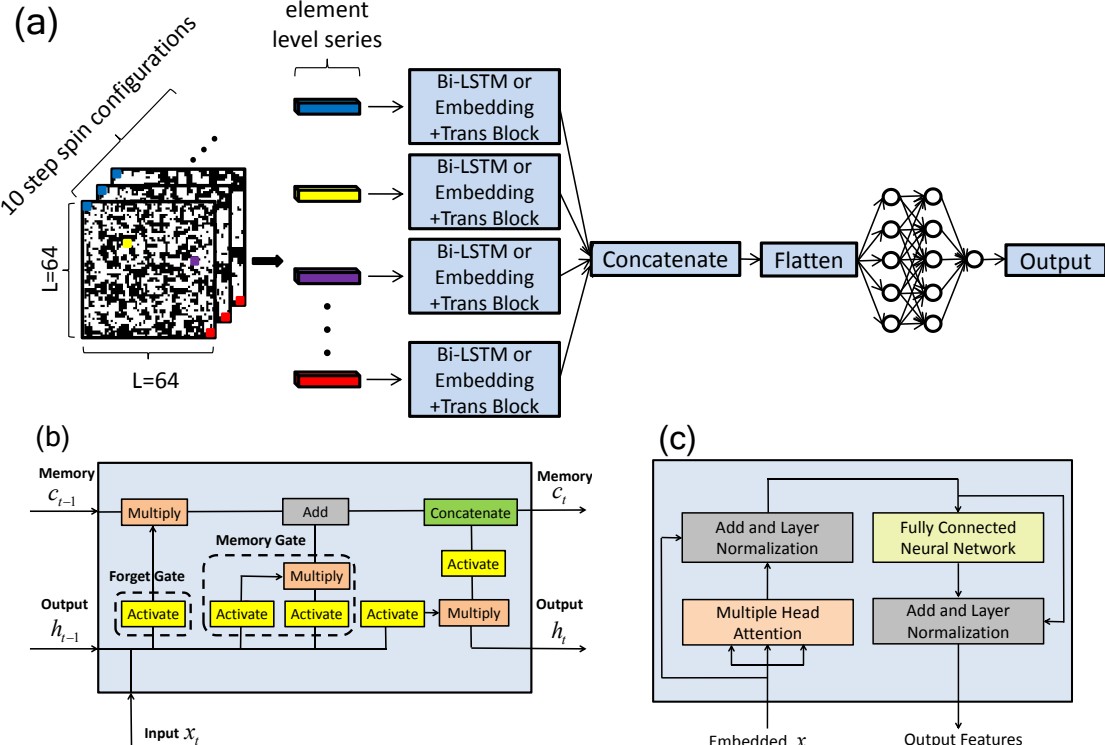

Figure 1: (a) Overall architecture of the deep learning model we used. (b) and (c) are the sub-architectures of the LSTM block and the Transformer block, respectively. The input to the model is 20 sequences obtained from MC simulation before reaching equilibrium, followed by 20 LSTM blocks or Transformer blocks to extract features from each sequential data, and finally fully connected layers are used to map the obtained features of the sequences into binary output to predict which phase the input data belongs to.

model, CNN can easily capture the spatial information of the configuration, such as vortexes in the XY model. However, in our task, the deep learning model needs to extract information from long sequences, and LSTM and Transformer are just suitable for such tasks [17, 18], so our deep learning model is followed by a Bi-LSTM block or a Transformer block after the input layer, then use FCN to process the extracted sequential information and output the probability of the binary classification. The structure of the deep learning model is shown in Fig.1 (a).

Despite RNN is often used to process sequential data, its simple internal structure makes it impossible to extract long-range correlated features in the data. On the other hand, LSTM has a more complex internal structure to better capture long-range correlated features. The architecture of LSTM is shown in Fig.1 (b). The internal structure of LSTM mainly consists of memory cells, forgetting gates, memory gates and output gates [17]. The function of memory cells ($c_T$) is to store important information in the input sequence ($x_T$) from $t = 0$ to $t = T$, the $T$ here is the time we feed the latest data into the LSTM blocks. The forgetting gate then judge whether there is any invalid information in the memory cell according to the input sequence ($x_T$) at time $t = T$ and the LSTM output ($h_{T-1}$) at time $t = T - 1$, and then set the vector value of the invalid information to be 0. The memory gate will judge what information needs to be added to the memory cell according to $h_{T-1}$ and $x_T$. The output gate then combines the information of $c_T$, $h_{T-1}$ and $x_T$ to determine the output of LSTM at time $t = T$.

Sometimes LSTM may fail in extracting long-range related features of the sequence because the sequence's elements has to be read one by one. This will overflow the memory cell information if the sequence is too long or leads to zeros in the gradient during backward propagation in the learning process. On the other hand, Transformer can extract all ranges of correlations in the sequence since it reads the elements of the entire sequence at one time. The correlation between all the elements of the sequence is then learnt through a self-attention layer and more advanced features can be extracted. The architecture of Transformer is shown in Fig.1(c) [18]. The self-attention layer is a key part of the Transformer. In principle, it can extract infinitely long-range correlated features.

Unlike Bi-LSTM, each sequence needs to be encoded into a feature space through the embedding layer before the sequential data is fed into the Transformer block, otherwise the self-attention layer in the Transformer block can not work efficiently. The embedding layer can be any encoder network in deep learning, for example, we use FCN containing one hidden layer with 50 neurons as the embedding layer. Usually, the positional information of each element in the sequence will also be added to the embedding layer after encoding into the space of the same dimension [18–20]. However, we found that this positional encoding is not necessary for our task as demonstrated in the Appendix A.1 that shuffling the order of the elements in the sequence does not have significant effect on the performance of the model. This is not surprising because our sequential data is obtained from Markov chain Monte Carlo simulation which is a no-memory process, whereas in tasks of natural language processing or computer vision, the position of a word in the sentence or the position of a pixel in an image carry important information.

In the following, we applied the above machine learning scheme to the classical and quantum many-body systems. For classical spin models with $N = L \times L$ sites, we used the spin configurations of each MC step as input data, which are $L \times L$ matrices. Here, we define a single MC step as an update that attempted to flip the spin on all the sites once. For the Hubbard model with $N$ sites, we used the Green functions as input, which are $N \times N$ matrices. An MC step here is referring to an update that attempted to flip 10% of the auxiliary fields. To let the deep learning model only focus on the information of the input data in the time dimension instead of the pattern in the space, we did not use the entire matrix as input. For each input sample, we randomly choose 20 elements in the matrix and pick the simulation result of these matrix elements in $m$ MC steps to form a tensor of the shape $(20, m)$, as shown in Fig. 1(a). We showed in the Appendix A.2 that selecting more elements does not improve the performance of the deep learning models significantly.

## 3 Machine performance in phase transition detection from data before equilibrium

The Ising model on a square lattice is a pedagogical model capturing the physics of a classical phase transition in condensed matter. The model describes spin-1/2 particles in a lattice system where each spin interacts with its nearest neighbours. The Hamiltonian of the Ising model is given by

$$H = -J \sum_{\langle i,j \rangle} \sigma_i \sigma_j, \tag{1}$$

where $\sigma_i \in \{-1,1\}$ denotes spin-down and spin-up respectively and the sum is over all the nearest neighbours. $J$ characterises the coupling strength between two nearest spins and is taken to be $J = 1$ in the following.

In an infinite size square lattice, the system exhibits a phase transition between the paramagnetic phase and the ferromagnetic phase at a temperature of $T_c = 2/\log(1 + \sqrt{2}) \approx 2.269$ [21]. When the temperature is close to zero, the interactions between the spins dominate and all the spins tend to align in the same direction. The system is in the ferromagnetic phase with an average magnetization $M \in \{-1,1\}$. When the temperature is much higher than $T_c$, the direction of the spins becomes random due to thermal fluctuations. The average magnetization of the system is approximately equal to zero and the system is in the paramagnetic phase. Given a randomly initialized spin configuration, one can simulate how this spin configuration reaches one of the two phases at different temperatures in equilibrium step by step using MC method.

Spin configurations of the Ising model at equilibrium have very obvious difference between ferromagnetic phase and paramagnetic phase. Previous work has shown that after supervised training of an FCN or an CNN using the equilibrium data, the neural network can easily locate the phase transition temperature of the Ising model [10]. However, for the Ising model, a randomly initialized spin configuration typically requires 500 MC steps to reach its equilibrium configuration at a given temperature. In more complex models, the time and computational resources required for the MC simulation to reach equilibrium will even be more. In the following, we explored whether the neural network can also accurately determine the phase transition temperature if it is trained with spin configurations from only the first few MC steps that are far before equilibrium is reached.

We used MC data obtained in the temperature range $T \in ([0,1] \cup [4,5])$ as the training set. The system size of the Ising model on a square lattice is $L = 256$. Five hundreds raw samples were generated in each temperature range, therefore we have a total of 1000 raw samples. As mentioned in Sec. 2, each input data of our deep learning model comes from the spin configurations of 20 randomly selected sites. For each raw sample, we repeatedly selected 20 sites randomly to obtain multiple training samples. Altogether, we obtained 10000 training samples from the MC raw data.

Figure 2 shows the output of the trained neural networks when fed with testing data from full temperature range. We have 30 samples for each temperature in the testing set and the error is calculated by the standard error of the deep learning model's output when fed with the corresponding testing samples. We tested the performance of FCN, CNN, Bi-LSTM and Transformer with different MC steps $m \in \{10, 20, 30, 40\}$. The machine outputs are fitted to a hyperbolic tangent function of the form $\theta_1 \tanh(\theta_2 x + \theta_3) + \theta_4$, where $\theta_1$, $\theta_2$, $\theta_3$, and $\theta_4$ are the fitting parameters, and the fitting curves are shown as the solid lines in the plots. From Fig. 2 (a) and (b) respectively, we found that Bi-LSTM and Transformer can accurately predict the transition temperature of the model to be $T_c \approx 2.269$ as determined by an output value of 0.5 (indicated by the horizontal dashed line in the figures) from the machine. The performance of the two models is insensitive to the chosen two temperature ranges where the training samples are taken from (see Appendix A.3). We also found that the predicted $T_c$ is not sensitive to the system size. Even if we go down to $L = 8$, where the 20 randomly selected sites consists of 30% of the total number of sites in the system, we can still obtain a reliable estimation of the transition temperature (see Appendix A.4). On the other hand, CNN can only predict that $T_c$ is around 2.269 and the results are also more sensitive to the number of MC steps (Fig.

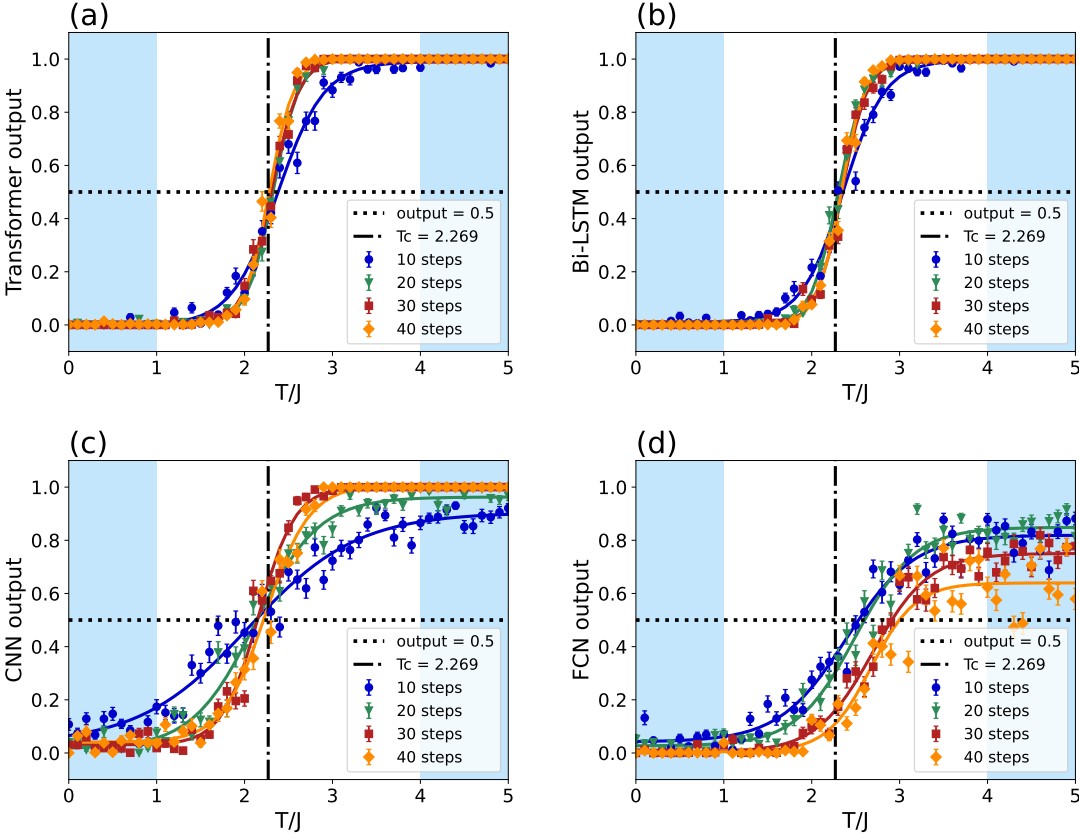

Figure 2: Testing result for Ising model on square lattice using (a) transformer, (b) Bi-LSTM, (c) CNN, (d) FCN. The light blue area represents the temperature range of our training data. Vertical black dashed line indicates the theoretically predicted transition temperature $T_c$. Solid lines show the tanh fit of the output data, and their intersects with the horizontal dashed line give the corresponding machine predicted $T_c$. Transformer and Bi-LSTM accurately predicted the phase transition temperature $T_c \approx 2.269$. The performance of CNN is relatively unstable. Even if the input data contains more MC steps, the predicted $T_c$ by CNN still has a relatively large error. FCN performed the worst, not only failed to predict the phase transition point, but also for the test samples within the training temperature range. FCN was unable to correctly classify the samples with high confidence.

2(c)). The performance of FCN is even worse. The predicted $T_c$ is significantly larger than the expected value (Fig. 2(d)). Besides, FCN is also less confident in classifying the two phases as for the test samples far away from $T_c$, its outputs do not reach 0 or 1. It is worth noting that in order to fairly evaluate the performance of each deep learning model, we controlled the number of parameters of the model to be in the same order of magnitude.

Figure 3 shows the predicted transition temperature by the four deep learning models using various number of MC steps $m$ in the input training data. Specifically, for each value of $m$, we trained the model 10 times. After each training, we fitted the machine output with the hyperbolic tangent function mentioned above. The temperature that corresponds to a fitted value of 0.5 is taken as the $T_c$ predicted by the model, and the error bar in the plot represents the standard error of the extracted $T_c$ from each training. For FCN and CNN, the predicted $T_c$ is unavailable when $m$ is too small since the tanh failed to fit the output data. From the figure,

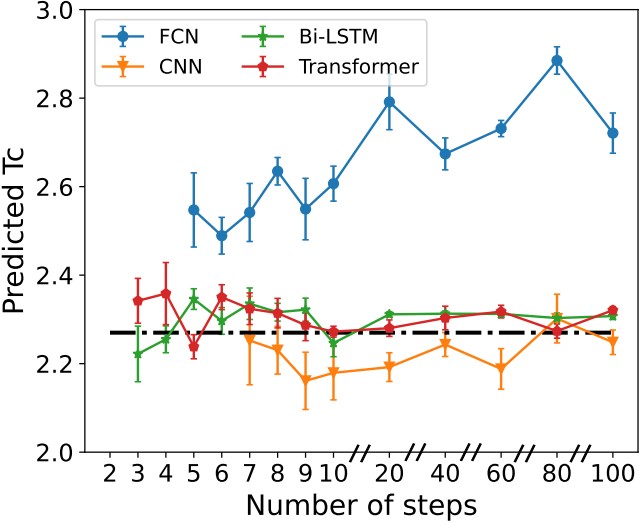

Figure 3: Machine predicted value of the phase transition temperature $T_c$ in the classical Ising model on a square lattice using different MC steps. The lines joining the data points are just guides for the eyes. In general, the transition temperature predicted by Transformer and Bi-LSTM better agrees with the theoretical value (horizontal black dashed line) than that predicted by CNN while The $T_c$ predicted by FCN is significantly larger than the theoretical value .

we find that except for FCN, the predicted value of $T_c$ from all the models are reasonably close to the expected value 2.269. However, the error in $T_c$ predicted by CNN is in general greater than that of Bi-LSTM and Transformer, which shows CNN is less stable than the other two models on this task, while Bi-LSTM and Transformer perform similarly well. We also extracted the predicted $T_c$ by performing a linear fitting to the output as shown in the Appendix B, and the same conclusion can be drawn.

We are not surprised by the poor performance of FCN as FCN has a natural disadvantage when dealing with sequential data. Suppose there is an FCN model with sequential element input, the output $z$ of the first neuron in the first hidden layer is given by

$$z = \sum_i w_i x_i , \tag{2}$$

where $w_i$ is the weight of each input element $x_i$. After the FCN model is trained, its weights $w_i$ will be fixed, and these weights will only depend on the position of the elements in the sequence. In other words, the weight of the first element of all input sequences will be the same. However, in our training data, the information contained in the elements at the same position in different sequences as well as their importance can be different in general.

The CNN model can solve the above problems by increasing the kernel number, while LSTM and Transformer convert the weight into an input-related function through a complex model architecture. Specifically, the hidden layer output of LSTM and Transformer become

$$z = \sum_i f(x_i) x_i , \tag{3}$$

where $f(x_i)$ is different for different deep learning models. In our task, $f(x_i)$ can help the deep learning model to better locate the key input data, and extract features from this key information to better complete the binary classification task. In fact, Transformer and Bi-LSTM have

very similar performances as we discussed above. This is because the input sequences' length, which is $m \in [2, 100]$, is within the storing capacity of the memory cell in Bi-LSTM and thus the full sequential information can be retained, making the advantage of Transformer not obvious.

# 4 Generalizability to other models and raw data source in time domain

We also applied our scheme to more complicated condensed matter models to test the robustness of our method. The models are the Ising model on a honeycomb lattice and on a triangular lattice, the XY model on a square lattice and the quantum Hubbard model on a honeycomb lattice. The system size of all classical models is $N = 256 \times 256$, while that of the Hubbard model is $N = 12 \times 12$.

The Hamiltonian of the XY model is given by

$$H = -J \sum_{\langle i,j \rangle} \cos(\sigma_i - \sigma_j), \tag{4}$$

where $J$ is the interaction strength between two nearest spins, $\sigma_i \in (0, 2\pi]$ represents the spin angle on the lattice's plane at the site $i$. Unlike the Ising model discussed in the previous section, the phase transition occurring at $T_c = 0.89$ in the XY model is Kosterlitz–Thouless (KT) type [22]. Above $T_c$, the spin correlation decays exponentially while it shows a power-law decaying behavior at temperatures below $T_c$. Vortexes and anti-vortexes with winding numbers equal to 1 and −1 respectively are formed in the system [23]. At low temperatures, the vortex and anti-vortex are tight to each other and tend to annihilate to minimize the system's energy. The phase transition is associated with the unbinding of the vortex-anti-vortex pairs at the critical temperature when the temperature increases. Intuitively, we shall expect these non-local spatial feature needs to be obtained through CNN using the entire spin configuration of the system. However, as discussed in Sec. 2, the input data used in our method is element level series in the simulation time domain. It will be interesting to test whether the deep learning model can still extract relevant information about the phases and accurately determine the KT phase transition.

The Hubbard model, on the other hand, is a quantum model whose Hamiltonian is given by

$$H = -t \sum_{\langle i,j \rangle} (c_{i\uparrow}^\dagger c_{j\uparrow} + c_{i\downarrow}^\dagger c_{j\downarrow} + h.c.) + U \sum_i n_{i\uparrow} n_{i\downarrow} - \mu \sum_i (n_{i\uparrow} + n_{i\downarrow}), \tag{5}$$

where $c_{i\sigma}^\dagger (c_{j\sigma})$ is the creation (annihilation) fermion operator of spin $\sigma = \{\uparrow, \downarrow\}$ at site $i$, $n_{i\sigma} = c_{i\sigma}^\dagger c_{i\sigma}$ is the number operator, $t$ is the nearest neighbor hopping amplitude, $U$ characterises the Coulomb interaction strength between two electrons of opposite spins at the same site, $\mu$ is the chemical potential. In this article, we considered $\mu = 0$, which corresponds to the case of half-filling. For $U > 0$, the system exhibits a quantum phase transition from the semi-metal phase to the antiferromagnetic insulating phase at $U_c \approx 3.9$ [24] as $U$ increases. Since the model is quantum in nature, input data is generated from quantum MC in which the raw output are the Green functions. We would like to investigate whether our method can be applied to this type of source data. Existing work has proven that Green functions after the equilibrium is reached in the quantum MC simulation are suitable input data for deep learning models to learn about the phase transition point [12]. As a data source for the deep learning

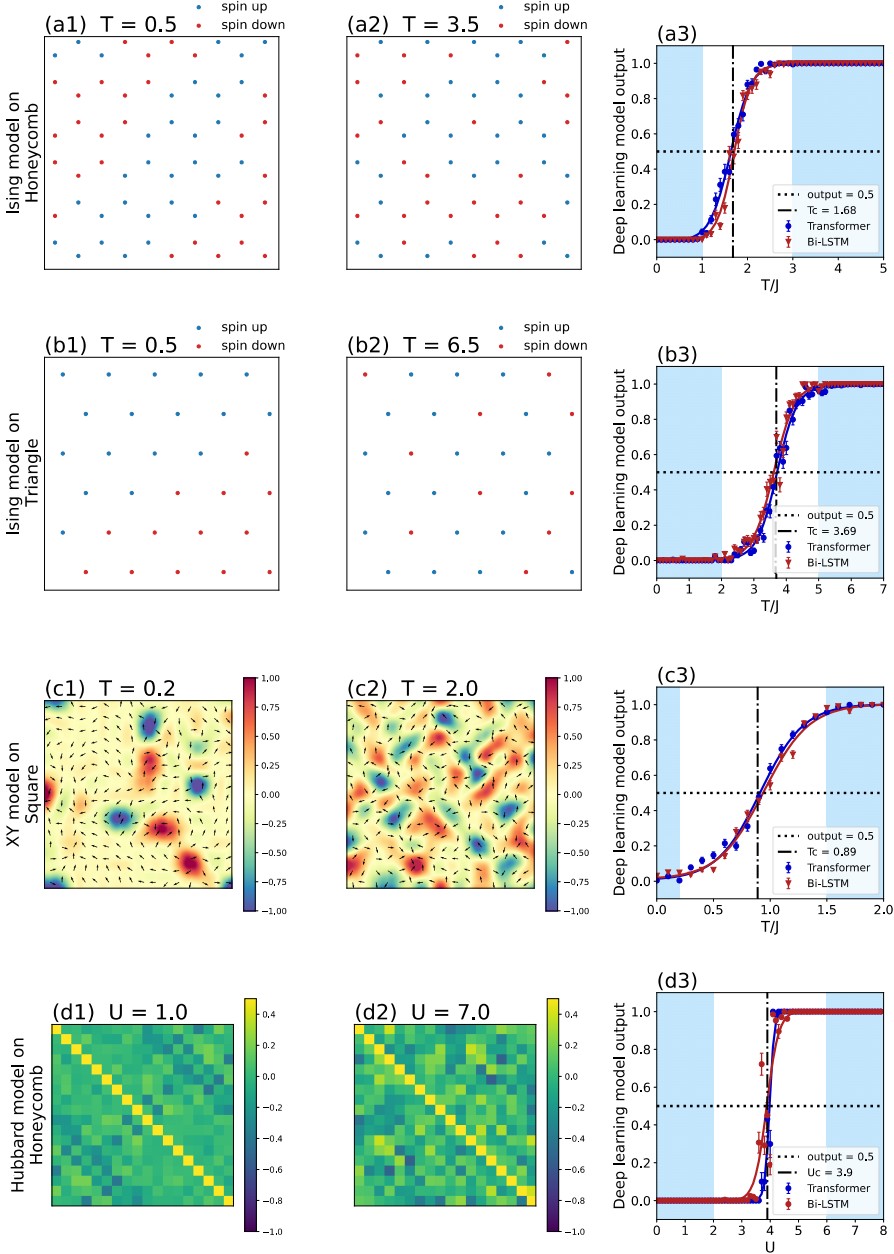

Figure 4: Figures (a1)-(a2), (b1)-(b2) and (c1)-(c2) are the spin configurations at the tenth MC steps in the Ising model on honeycomb lattice, Ising model on triangle lattice and XY model on a square lattice, respectively, away from the phase transition temperature. Figure (d1)-(d2) are Green functions of the Hubbard model away from the quantum phase transition point. In the Ising models, the blue dots represent up spin and the red dots represent down spin. In the XY model, arrows represent spin orientation and color scale represents the magnitude of the local winding number. In the Hubbard model, the color scale represents magnitude of the Green function's matrix element. Figure (a3), (b3), (c3) and (d3) shows the output of the Transformer and Bi-LSTM in the corresponding condensed matter models respectively. Transformer and Bi-LSTM in the three classical models perform similarly to the Ising model on a square lattice. However, their performance in the Hubbard model is significantly different. The output value of the deep learning model is unstable and it shows a step-like abrupt change in the close vicinity of the transition point.

model, the physical interpretation of the Green function is very different from the spin configuration. For example, the spin configuration represents the spatial feature of the system in the real space, while the Green function represents the correlation between the creation and annihilation of the electron from different sites in the Hubbard model. During the training, we use the time-series of randomly picked elements from the Green functions. In the case of the spin configuration, the translational symmetries in the model help preserve the universality of the training data. However, this is not the case for the Green function, the time variation of the correlators highly depends on which element in the Green function we picked. This may be an adverse factor when we train the model using the Green function.

Figure 4 shows the results of the above mentioned models using Bi-LSTM and Transformer. The error bar in the plot is obtained in the same way as that in Fig. 2. Our deep learning model performs well on all the three classical spin models (Fig. 4(a)-(c)). In the regions far from the phase transition point, the deep learning models can distinguish the two phases with a high level of confidence, while near the phase transition point, the deep learning model has a very smooth change due to confusion. When the output of the deep learning model is equal to 0.5, its corresponding temperature $T_c$ is approximately consistent with those obtained in the literature. The performance of our proposed deep learning model in the Ising models are in line with our expectations. Since the only difference among these models is in the geometry of their lattices and only the nearest-neighbor ferromagnetic interactions are present in the models, one shall expect the same type of phase transition between the FM and PM to take place. However, to our surprise, the deep learning model also performs well on the XY model where the input data we used to train the model contains no spatial information. Even the model cannot capture any information about the topological quantities in the data, the model can still classify the two phases well. This suggests the sequential information present in the MC steps before equilibrium may be related to the topological character of the XY model at equilibrium. Unfortunately, it is hard to interpret what features have the deep learning models learnt from the sequences due to the complexity of the network itself.

In the case of the Hubbard model (Fig. 4(d)), the input data to the deep learning model is a sequence of length 100, which is about 0.01% of the quantum Monte Carlo(QMC) steps required to reach equilibrium. We found that the output of the deep learning models does not change gradually as in the classical models near the phase transition point, but exhibit a significant increase from 0 to 1 as the interaction strength $U$ increases, as shown in Fig. 4 (d3). Moreover, the output of the deep learning models is unstable in the vicinity of the quantum phase transition. For example, samples in the semi-metal phase are sometimes classified as an antiferromagnetic insulator phase with high confidence. We can be sure that this difference is not because each matrix element in the Green function contains much less information than the spin configuration of the classical models. As we can see from the figure, most samples near the transition are correctly classified with very high confidence. If the information in the input data is not enough, the deep learning model will output a probability that is much greater than 0 and much less than 1 due to confusion. Instead, the large fluctuation of the output is mainly contributed by two factors. Firstly, unlike the spin configuration in classical models, the input using the Green function does not obey translational symmetry, so the behavior of the sequence depends a lot on the choice of the element's location. This will easily lead to misjudgement if we pick the element with very slight difference in the two phases. Secondly, the elements in the Green function fluctuate a lot, especially in the vicinity of the phase transition point due to quantum fluctuation. The large fluctuation in the input data further hinders the deep learning model from classifying the phases correctly. To improve, we find that the fluctuation drops slightly if we increase the number of quantum MC steps $m$ (see

Table 1: Time cost in determining the phase transition point in the Ising and XY models on a square lattice, and Hubbard model on a honeycomb lattice by three different methods. The blue and teal color elements correspond to the time spent in data collection and data processing, respectively. Zero refers to the case where the time spent is less than the time measurable by the computer. For medium to large system sizes, our proposed method (bottom row) significantly reduces the overall time spent.

| Model (size) | Ising (8 × 8) | Ising (64 × 64) | Ising (128 × 128) | XY (128 × 128) | Hubbard (9 × 9) |
|---|---|---|---|---|---|
| Order parameters from equilibrium data | 5 min<br>0 | 5 hr<br>0 | 18 hr<br>0 | 14 hr<br>0 | 733 hr<br>6 s |
| Supervised learning (FCN) from equilibrium data | 2.5 hr<br>3 min | 150 hr<br>3 min | 550 hr<br>3 min | 782 hr<br>3 min | 9900 hr<br>3 min |
| Transformer/Bi-LSTM using before equilibrium data | 1 min<br>15 min | 0.95 hr<br>15 min | 3.9 hr<br>15 min | 5.7 hr<br>15 min | 35 hr<br>15 min |

Appendix C).

Table 1 shows the approximated time cost in determining the phase boundary using our scheme as compared to the traditional method of estimating the transition from order parameters and to the supervised learning method (with FCN containing one hidden layer with 100 neurons) [10] using MC or quantum MC samples after reaching equilibrium. The time cost is evaluated on a computer with Intel i5 CPU and NVIDIA GTX1660 GPU. We presented the time cost for data collection, which refers to generating the spin configurations or Green functions (see Appendix D for the details), and that in data processing, which refers to training the deep learning models or calculating the order parameters from the spin configurations or Green functions. It can be seen from the table that our method takes much less time in data collection than the other two methods while the time taken for data processing is relatively longer due to the complexity in Transformer and Bi-LSTM themselves. However, this longer time spent in data processing becomes more negligible in large systems when the time cost in collecting equilibrium data becomes much more expensive. For example, our method gains an overall time reduction of 99% and 77% as compared to FCN and order parameter calculations using equilibrium configurations in the Ising model of system size 128 × 128. Observing that most of the time spent in collecting the equilibrium data instead of training the model, we expect such a significant speedup will also be possible in other supervised learning models that require a similar amount of training data.

## 5   Conclusion

In this work, we showed that deep learning models can rapidly classify phases in various condensed matter models using MC data before equilibrium and locate the critical points with high accuracy. Among the deep learning models we have investigated, the performances of Bi-LSTM and Transformer are found to be the best in probing the phase transition points. Both models are mainly constructed to extract features of time-domain data. Unlike the CNN counterparts, their success relies on the efficient feature extraction from the long sequential input

data, which have not been explored in the mission of phase transition detection in previous studies.

We also investigated the generalizability of our method to the Ising model on a honeycomb lattice and a triangular lattice, the XY model, which undergoes a KT transition, and the Hubbard model where the input training data comes from the Green functions generated by quantum MC. Bi-LSTM and Transformer can determine the critical points of these condensed matter models accurately. The results evidence that our proposed method is robust in detecting various types of phase transitions in condensed matter models and in using different types of source data.

We would like to remark that the data generated by the Markov Chain Monte Carlo simulation in this study contains no time order. Bi-LSTM and Transformer do not extract features in time order, but only information on the sequences' elements. For future works, it will be worth to apply the method proposed here to real time-ordered data, such as molecular dynamics simulated data, to study the dynamics of disordered systems or glass transitions.

The data and the codes for generating the plots in this article are available at https://github.com/ParcoDing/Rapid-detection.

# Acknowledgements

We thank the two anonymous reviewers for their helpful suggestions and comments to improve the earlier version of this paper. We acknowledge financial support from National Natural Science Foundation of China (Grant No. 12005179), Harbin Institute of Technology Shenzhen (Grant No. ZX20210478, Grant No. X20220001) and City University of Hong Kong (Grant No. 9610438).

*

# A   Analysis of model performance

In this section, we varied the training sample parameters to investigate the performance of our deep learning models in detecting the phase transitions in the condensed matter models. These parameters include the range of the driving parameters where the samples are taken from, the number of randomly selected sites, the number of MC steps $m$. We also discussed the system size effect and the effect in shuffling the elements of the sequential input.

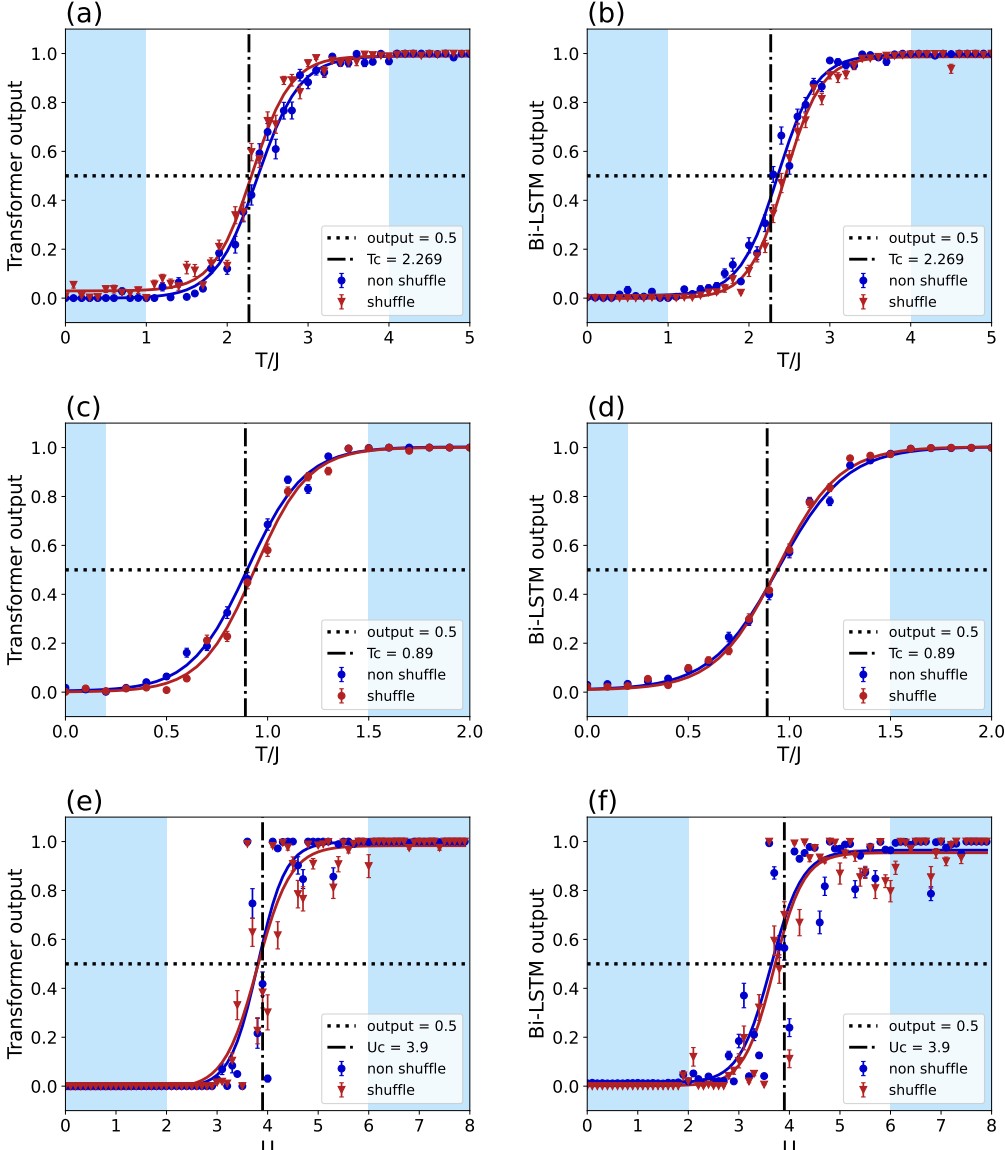

Figure 5: Performance of Transformer and Bi-LSTM on the Ising model on square lattice ((a) and (b)), XY model on square lattice ((c) and (d)) and Hubbard model on honeycomb lattice ((e) and (f)). Here, the MC steps $m = 10$ is used. The light blue area represents the temperature range of our training data. Solid lines represent the tanh fit to the output data. Shuffling the data in the simulation time dimension does not have significant effect on the performance of Transformer and Bi-LSTM.

## A.1 Shuffle time dimension input data

We experimented whether using positional embedding or shuffling the data in the time dimension will change the performance of the deep learning models. Figure 5 (a-d) shows the outputs of the deep learning models for the Ising model and the XY model on a $N = 32 \times 32$ square lattice, respectively. We can see that the performance of Transformer and Bi-LSTM are not affected significantly by the shuffling. One may understand this by the fact that the MC simulation is a Markov chain process, where the action of each step in the simulation is independent of the previous steps. From 5 (e) and (f), we also demonstrated in the Hubbard model on a $N = 6 \times 6$ honeycomb lattice that the deep learning models perform similarly no

matter if the data in the time dimension is shuffled or not.

## A.2   Varying number of randomly selected elements

Figure 6 shows how the output of the LSTM and Transformer changes when the number of randomly selected sites (elements) varies for the Ising model on a square lattice. Here we kept the number of MC steps fixed to 10. The plots show that the output of the deep learning models have larger fluctuations as the number of elements decreases in both cases of $L = 16$ and $L = 32$. Even for temperatures far away from the transition temperature, the deep learning models have less confidence in its prediction (the output is not either 1 or 0). The situation improves if the number of elements increases to 20 or above. However, further increasing the number of elements beyond 20 does not significantly increase the accuracy of the predicted $T_c$. Considering the complexity of the deep learning models, the amount of memory usage and the implementation of the model, using 20 elements in the input is a reasonable choice.

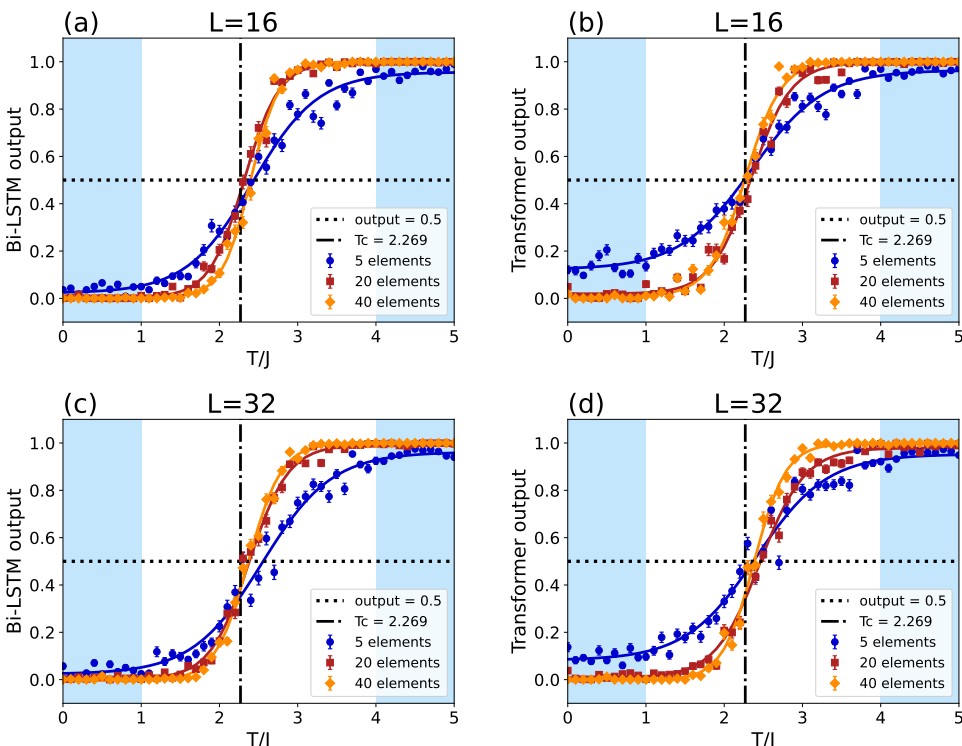

Figure 6: Outputs of the deep learning models for an $L = 16$ (left) and $L = 32$ (right) Ising model on a square lattice with different number of randomly selected sites (elements). Solid lines show the corresponding tanh fit to the output data.

## A.3   Varying the temperature range of the training samples

We used the first 10 Monte Carlo steps and 20 randomly selected sites in the Ising model on a square lattice prepared from different temperature ranges to feed the Bi-LSTM and Transformer. As shown in Fig. 7, Bi-LSTM and Transformer perform very similarly in the four different sets of training temperature ranges.

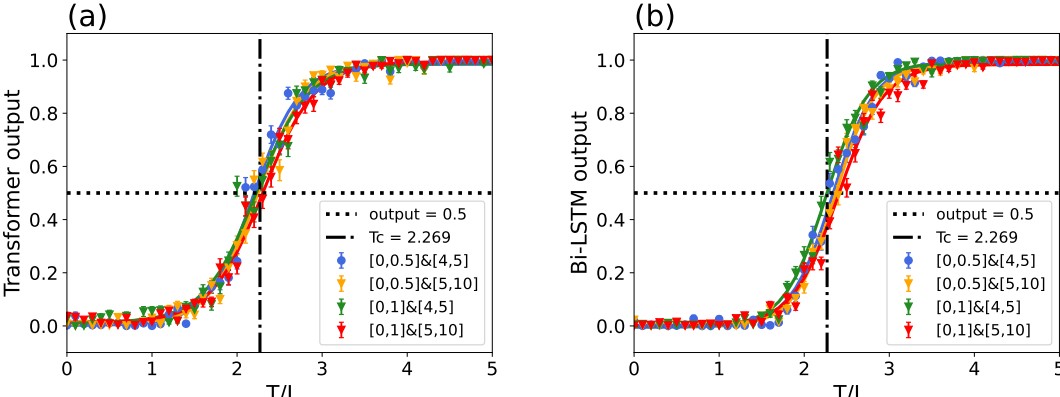

Figure 7: (a) and (b) show the output of Transformer and Bi-LSTM on Ising model on square lattice, respectively. The legend indicates the range of temperatures in which the training samples are taken from. It can been seen that the performance of the machine is not sensitive to the training range.

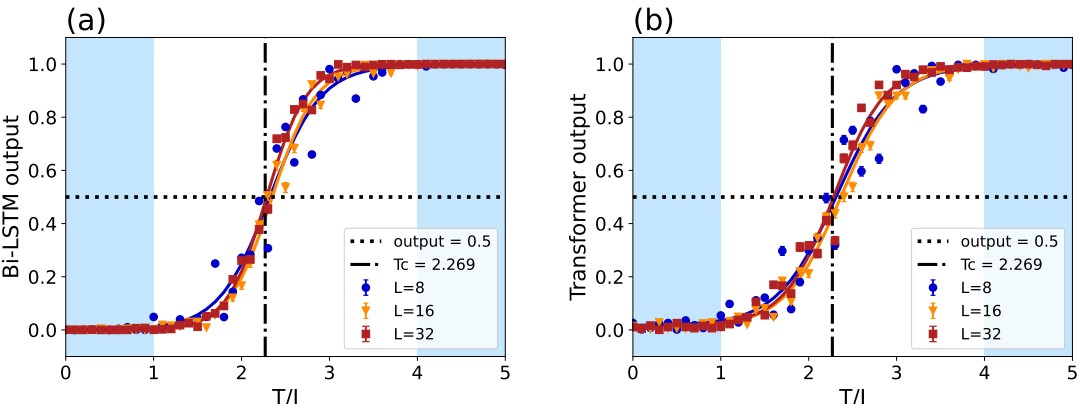

Figure 8: (a) and (b) show the output from Bi-LSTM and Transformer of the Ising model on square lattice with different system sizes, respectively. Solid lines indicate the fitting of the data points to the hyperbolic tangent function presented in the main text. The $T_c$, as indicated by the intercept of the fitted lines and the horizontal dashed line, agrees well with the theoretical value (vertical dashed line).

## A.4 System size effect

In Fig. 8, we investigated the system size dependence in applying our scheme to extract the phase transition temperature in the Ising model on a square lattice. Similar to the case of $L = 256$ shown in Fig. 2, the number of MC steps used here is $m = 10$ and 20 sites are randomly selected to form the training data. We find that the extracted $T_c$ is insensitive to the system size and agree well with the theoretically predicted value, despite the fact that fluctuation in the machine's output increases around the transition temperature when the system size decreases.

# B    Extracting the phase transition points from linear fitting

Besides fitting the deep learning models' output to a tanh function shown in the main text, we also performed a linear fitting to estimate the transition temperature of the $L = 256$ Ising square lattice and the result for various MC steps $m$ is shown in Fig.9. Here the linear regression is carrying out for data in the temperature range 0.1 to 0.9. The conclusion that Bi-LSTM and Transformer generally performs better than the FCN and CNN remains true. From the plot, we can see that when $m < 7$, the output from both FCN and CNN fluctuates a lot, and the predicted transition points are also far away from the theoretically predicted value (dashed horizontal line). When $m \geq 7$, the result of CNN begins to converge to the theoretical value, while FCN still performs poorly. In contrast, the transition temperature located by Bi-LSTM and Transformer converges to around the expected value much faster with respect to the number of steps.

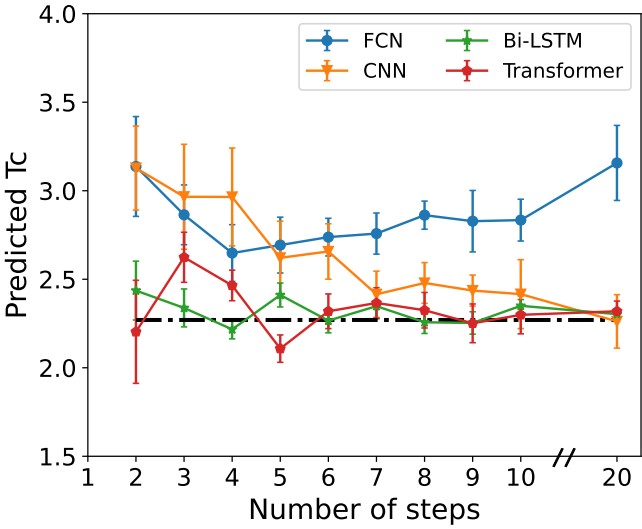

Figure 9: The transition temperature of the Ising square lattice predicted by four different deep learning models with various MC steps. The settings are the same as that shown in Fig. 3 except that the transition temperature is extracted from fitting the deep learning models' output in $T \in [0.1, 0.9]$ linearly.

# C    On the machine output's fluctuation in the Hubbard model

In the main text, we showed that although our method can accurately locate the transition point $U_c$ in the Hubbard model on a honeycomb lattice, the output of the deep learning model fluctuates around the transition point. We further investigated whether increasing the length $m$ or the number of Green function elements in the input sequences can reduce the fluctuation. Fig. 10 shows the difference between the output of the deep learning model and their corresponding tanh fit. The larger the difference is, the stronger the output fluctuation of the deep learning model is. From Fig.10(a) and (b), we observed that the output fluctuation increases when coming closer to the transition point. The result shows that increasing the sequence length can reduce the difference mildly. However, we cannot see obvious improvement when we increase the number of input sequences in Fig. 10(c) and (d).

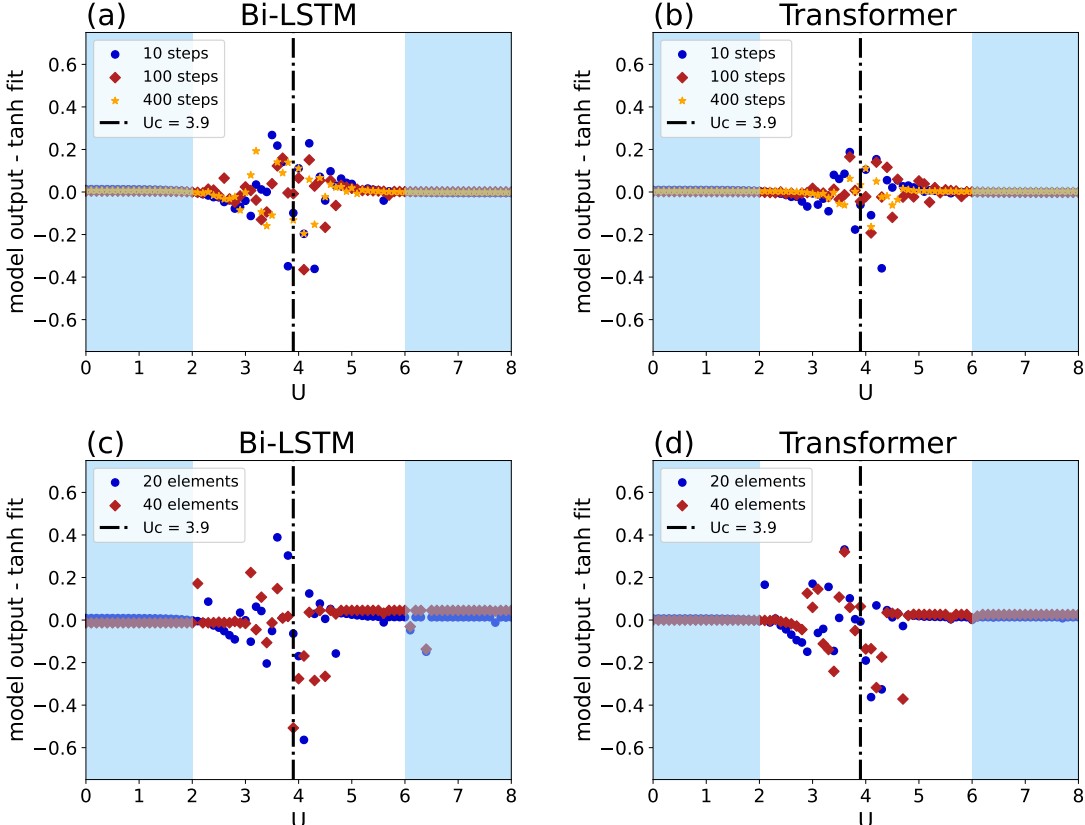

Figure 10: Fluctuations in the output of the deep learning models as measured by the differences between the output value and the tanh fit in the Hubbard model. Though slight improvement can be found when increasing the number of MC steps, there is no significant reduction in the fluctuation in increasing the number of selected Green function elements.

# D  Sample collection details for Table 1

For the method on calculating the order parameter in the Ising model, we selected 50 values of temperature in the range $T \in [0, 5]$ with an evenly spaced interval, and collected 10 MC samples for each temperature. Specifically, for each temperature, we first warm up the simulation with 500 MC steps, and then collect one sample every 100 steps, altogether simulated 1500 MC steps.

For the method on supervised learning phase transitions in the Ising model from equilibrium spin configurations using FCN, to prepare the training set, we randomly selected 1000 temperature values in the range $T \in ([0, 1] \cup [4, 5])$ and the MC samples are collected in the same way as mentioned above. A total of 10000 training samples are obtained. The test set is collected in the same way but with 50 evenly spaced values of temperature in the range $T \in [0, 5]$.

For our method on Transformer/Bi-LSTM with spin configurations before equilibrium, we again randomly selected 1000 temperature values in the range $T \in ([0, 1] \cup [4, 5])$, but now we only need 10 MC steps per temperature value from 20 randomly selected sites. A total of 10,000 training samples are obtained. The test set is prepared in the same way in the range

$T \in [0, 5]$ with an interval 0.1.

For the Hubbard model, the data collection is similar to that for the Ising model. Each equilibrium Green function is obtained by averaging the Green functions of multiple quantum MC steps (we do not need to consider the sign problem in the case of half-filling). In the order parameter calculations and learning with FCN using equilibrium Green functions, we first warm up the simulation for 100 steps and then collect the Green functions with 1000 steps to obtain an equilibrium sample. For our method, we do not need a warm-up session or obtain equilibrium samples through multiple steps, we can directly collect the non-equilibrium Green functions of the first 10 quantum MC steps as our data.

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
