# Peer review of "Rapid detection of phase transitions from Monte Carlo samples before equilibrium"

_SciPost Physics, doi:SciPost Phys. 13, 057 (2022)_

## Round 1 · Referee Report · Anonymous (Referee 1) · 2022-4-27

Strengths

1- The authors introduce a way to reduce the calculation time of Monte Carlo methods by using different neural network architectures which leads to more efficient calculations.
2- very good that the authors check if the performance of the model is independent of the selection of the training data
3- The authors check different models
4- good graphical representation of the results and the architecture

Weaknesses

1- sometimes unclear text structure. The authors repeat some information in the different paragraphs.
2- fit to find the phase transition from the output of the neural networks (see requested changes)
3- The concrete implementation/code of the neural network architectures is not publicly available.

Report

In the paper „Rapid detection of phase transitions from Monte Carlo samples before equilibrium“ by Ding et al. the authors use different machine learning techniques, namely a Transformer and a Recurrent Neural Network, to detect phase transitions from Monte Carlo samples before reaching equilibrium. They test their new technique on the Ising model, Hubbard Model and the XY model.
The authors give a good introduction into the field which address a broad audience. They motivate well the importance of their work. They give clear visualizations of the techniques and the data. Furthermore they tested their technique with different physical and machine learning models as well. In the appendix they show the stability for different training data/training ranges which is required for a sophisticated machine learning algorithm.
The contribution is of general interest since it can reduce the Monte Carlo steps needed to find phase transitions which reduces the amount of computational resources involved.
I recommend publication in SciPost Physics if the authors address the questions/changes.

Requested changes

1- In the end of section 1 (e.g. first paragraph on page 3) the authors describe in detail how the method works. In section 2 this is done again. I kindly ask to reduce the repetition here.
2- The data for figure 3 is generated by doing a linear regression on the data from figure 2 in the interval of 0.1 to 0.9. I would like to recommend to fit a tanh like function to the complete data to get a more stable point since the restriction of 0.1 and 0.9 as boundary is an arbitrary choice. Nevertheless I do understand that no model is available that suits the data from a theoretical point of view, however the fit of a tanh function appears often in such publications where a phase transition needs to be found from a supervised trained network. As an alternative it would be helpful to either see the linear fits in the main text or in the appendix. The authors could plot the fitted function instead of the guide to the eye in figure 2 and 4.
3- How do the authors calculate the errorbars in figure 2 and figure 4?
4- I do not completely understand paragraph 2 on page 5 even after reading the reference in the appendix. Why and how do the authors map the information in the different dimensions? Could the authors comment what they are doing here?
5- The authors comment that for the Ising model normally 1000 MC steps are required to reach equilibrium. To train the neural networks some training data is required. Could the authors directly compare (for example with an overall factor) how much less/more data the methods need to find the phase transition? How big is the actual reduction of computational requirements taking the complete progress into account.
6- Please review minor grammar mistakes and typos. (typos especially in the figures. for example fig 1: "menory" -> I think the authors intention is "memory" )
7- I would like to draw attention to the formal acceptance criteria which state that the publication of the code/notebooks to create the figures are required. I was not able to find if and where it is available. To check if the models are under or overfitting I would like to recommend to also publish the training progress/concrete implementation of the neural networks.

  • validity: high
  • significance: good
  • originality: high
  • clarity: good
  • formatting: excellent
  • grammar: good

Author:  Wing Chi Yu  on 2022-06-27  [id 2610]

(in reply to Report 1 on 2022-04-27)

We thank the referee for carefully reviewing our manuscript and his/her helpful suggestions. Please find below our responses to each of the corresponding points in the Referee's suggested changes.

#1: We thank the referee for drawing our attention to this redundancy. In the revised manuscript, we have removed the detailed description of the method in Section 1 and left the details to Section 2.

#2: We thank the referee for the suggestions. In the revised manuscript, we have replaced Fig. 3 with the transition temperature that is extracted by fitting the machine output with the function a.tanh(b. x +c)+d, where a,b,c,d are the fitting parameters. The fitted curves for m =10, 20, 30, 40 are plotted in Fig. 2 for visualisation. The original Fig. 3, where the Tc is extracted by linear fitting the data, is now moved to Appendix B for reader’s reference. In both cases, the conclusion that transformer and Bi-LSTM perform better remains unchanged. The solid lines in Fig. 4 are also replaced by the fitted tanh curves accordingly.

#3: For the testing set, we generated 30 spin configurations or Green functions from MC or QMC simulation respectively for each driving parameter value in the full considered range. The testing set is then fed to the trained deep learning model and the error bars are calculated by the standard error of the predicted output of the 30 samples. We have clarified this in the last paragraph of P.6 when mentioning Fig. 2, and in the second paragraph of P.11 when mentioning Fig. 4.

#4: Unlike Bi-LSTM, each sequence needs to be encoded into a feature space through the embedding layer before the sequential data is fed into the Transformer block, otherwise the self-attention layer in the Transformer block can not work efficiently. The embedding layer can be any encoder network in Deep Learning, for example, we used an FCN which contains one hidden layer with 50 neurons as the embedding layer in our case. Usually, in the task of natural language processing or computer vision, the positional information of each element in the sequence will also be added to the embedding layer after encoding into the space of the same dimension. However, we found that this positional encoding is not necessary for our task where we have demonstrated in Appendix A.1 that shuffling the order of the elements in the sequence does not significantly affect the performance of the model. This is not surprising because our sequential data is from Markov chain Monte Carlo simulation which is a no-memory process, however, in the tasks of natural language processing or computer vision, the position of a word or the position of a pixel carries important information. In the revised manuscript, we have modified the discussion in the second last paragraph of Section 2 to clarify the use of the embedding layers.

#5: In the revised manuscript, we have included Table 1 summarising the time cost in sample generation and in training the deep learning models in various condensed matter systems using our method as compared to the other two methods, namely by calculating order parameters and supervised learning (with FCN) in Ref. [10] from equilibrium MC samples in determining the transition point. The detailed data collection process is presented in Appendix D. In both machine learning approaches, the number of training samples used is the same. We can see from the table that for small systems (e.g. an 8x8 Ising square lattice), the total time cost of our method is about 3 times larger than that of the order parameter calculation method. Most of the time spent in our method is on training the deep learning model due to the model’s complexity itself. However, significant speed-up can be achieved by our method in large systems (e.g. when N=128x128) and more complicated systems such as the quantum Hubbard model where the time cost in collecting equilibrium samples becomes expensive. Specifically, for the 128x128 Ising model, 99% and 77% of the computational time is reduced when compared with the FCN supervised learning and order parameter calculation with equilibrium MC samples, respectively. We have added the corresponding discussion to the second paragraph below Table 1 on P.12.

#6: We thank the referee for carefully reading our manuscript and pointing us to the typos. We have corrected the typos in Fig. 1 and reviewed the spelling and grammatical accuracies of the text.

#7: We have made the raw data and codes publicly available and have included the link (https://github.com/ParcoDing/Rapid-detection) at the end of the revised manuscript.

---

## Round 1 · Referee Report · Anonymous (Referee 2) · 2022-4-29

Strengths

1- New deep learning based method that allows for phase transition probing with less computational effort than the traditional Monte Carlo approaches.
2- Versatility of the method, which works well on a variety of models with rather distinct phase transitions.
3- The authors discuss several "internal" parameters of the method that are important in controlling its performance.
4- Good contextualization of the work.
5- The figures are generally good, clear and informative.

Weaknesses

1- Absence of a performance comparison against other established methods.
2- Some details, knobs and performance of the methodology could have been explored/presented more thoroughly.
3- Raw Monte Carlo data and deep learning scripts are not available to the readers.
4- Several misspellings across the text and occasional unclear wording.

Report

In this manuscript the authors estimate the location of phase transitions of several model Hamiltonians by employing supervised machine learning algorithms tailored for time domain analysis. Unlike previous works, the Monte Carlo data fed to the algorithms was not collected at equilibrium, but was instead extracted from the initial portion of the Monte Carlo time series before equilibrium is reached. This procedure should decrease the computational cost needed for estimating critical temperatures and couplings of physical models. This work is original, technically sound, the obtained results are interesting and of practical relevance for the field. I suggest the acceptance of this manuscript provided the following concerns are addressed by the authors.

Requested changes

1- The authors state that this methodology allows for cheaper estimates of the position of phase transitions of model Hamiltonians. Despite this being a credible statement it would be useful for the readers to know how much can one save with this method. Even a semi-quantitative comparison of the cost of this method (for a fixed accuracy of the prediction), against the usual Monte Carlo estimate based on expectation values, and against the supervised learning methods using equilibrium samples (eg. Carrasquilla et al., Ref. [10]), would be useful.

2- It is unclear from the text if the Monte Carlo time series (for a given site) that is fed into the deep learning algorithms, is sequential (i.e., each Metropolis step/spin flip attempt corresponds to an element of that site time series), or if there is some sort of reblocking to decorrelate samples (eg., only after a sweep over all the spins do we add a new element to the time series of a given site). For the benefit of the reader this should be clarified in the text.

3- It is a well known fact that the estimate of the phase transition will depend on the size of the simulated system. With finite size scaling techniques one can account for that and obtain reliable thermodynamic limit estimates. The methodology presented in this manuscript should not be immune to this issue, since it originates from the cutting-off of correlations by the finite size of the system. In the manuscript there is no study/discussion of the change of the predicted Tc with the system size. The simulated classical models are large, so the Tc prediction should be rather close to the thermodynamic limit Tc. Still, I believe that the average reader would benefit from this issue being discussed in the text. In particular, it would be useful to have a sense of how small can one make our simulations and still obtain reliable results with this method (given that the authors are randomly selecting 20 sites to feed into the algorithm). Would problems arise if these random sites are too close to each other?

4- The authors could fit the data in Fig. 2 with a function that better describes the data (eg., Tanh[a.x-b]) instead of using a linear fit between 0.1 and 0.9. I believe this would enable a more controlled estimate of the transition and its associated error. If there is some strong reason for using a linear fit between 0.1 and 0.9 instead of a function better adapted to the data, then that could be addressed in the text. In any case, the fits could be shown in the appendix or at least made available online.

5- It is not clear from the text how are the error bars in Fig. 3 estimated. For the benefit of the reader please clarify this.

6- Fig. 3 is rather useful to assess the convergence of the different algorithms with the number of steps of the time series. However I find it could convey more information if for instance, more points at larger number of steps were included (notice that data for m=20,30,40 is included in Fig. 2). From the text it is unclear whether there is a finite optimal number of steps for making a prediction, or if one should instead take the number of steps towards infinity to controllably approach the exact result. The readers would benefit from such discussion.

7- The authors mention that they chose to randomly select 20 elements/sites because selecting more sites did not improve the deep learning model. It could be useful to have a figure conveying such an information, together with a discussion of why 20 sites is the right number. I wounder if that optimal value of 20 sites depends on the system size and/or model.

8- In the last paragraph before the conclusion the text reads: "...exhibit a significant increase from 0 to 1 as temperature increases, as shown in Fig. 4(d3).". I believe this to be a typo since the authors are discussing the Hubbard model, which has a quantum phase transition upon tuning of the Hubbard interaction.

9- The presented method gave a reasonable estimate of the critical coupling Uc of the honeycomb lattice Hubbard model, despite not as good as the estimates of Tc for the classical models. It is not entirely clear to me why is that so. I would nevertheless like to remark that with the supervised deep learning methods that use equilibrium samples this seems to also occur: application to the Hubbard model (Broekcker et al., Ref. [12]) seems to result in a biased estimate of the critical coupling, while the Tc of the classical models (Carrasquilla et al., Ref. [10]) are rather well estimated. Related to this the authors mention that "the large fluctuation of the output in the close vicinity of the transition point may attribute to the fact that the Green functions fluctuate a lot (as compared to the classical model) due to the quantum fluctuations when they are far from equilibrium, resulting in the misclassification of some test samples.". Can this issue be improved with a longer time series sequence? Else, does it help to use more elements of the Green's function to feed the algorithm? Please discuss these issues.

10- In appendix A the authors state that "not using positional embedding or shuffling the data in the time dimension will not change the performance of the deep learning model.". That is demonstrated for the square lattice Ising model. Could the authors discuss/demonstrate if this statement holds for the other models (namely the XY model and Hubbard model)?

11- I believe that the Temperature label on Fig. 4(b2) might be wrong since the Tc in that system is 3.69 and that label reads T=3.5. Please check this possible issue.

12- The community would benefit from the raw Monte Carlo data, the deep learning, the fitting and the figure-generating scripts being made available online.

13- There are several misspellings across the text that could hinder the understanding of the concepts being discussed in the manuscript. I suggest the authors go through the text again in order to fix these typos.

  • validity: high
  • significance: high
  • originality: high
  • clarity: good
  • formatting: good
  • grammar: good

Author:  Wing Chi Yu  on 2022-06-27  [id 2609]

(in reply to Report 2 on 2022-04-29)

We thank the referee for carefully reviewing our manuscript and his/her helpful suggestions. Please find below our responses to each of the corresponding points in the referee's requested changes.

#1: We thank the referee for the suggestion. In the revised manuscript, we included Table 1 summarising the time cost in sample generation and in training the deep learning models in various condensed matter systems using our method as compared to the other two methods, namely by calculating order parameters and supervised learning (FCN) in Ref. [10] from equilibrium MC samples in determining the transition point. The detailed data collection process is presented in Appendix D. In both machine learning approaches, the number of training samples used is the same. We can see from the table that for small systems (e.g. an 8x8 Ising square lattice), the total time cost of our method is about 3 times larger than that of the order parameter calculation method. Most of the time spent in our method is on training the deep learning model due to the model’s complexity itself. However, significant speed-up can be achieved by our method in large systems (e.g. when N=128x128) and more complicated systems such as the quantum Hubbard model where the time cost in collecting equilibrium samples becomes expensive. Specifically, for the 128x128 Ising model, 99% and 77% of the computational time is reduced when compared with the FCN supervised learning and order parameter calculation with equilibrium MC samples, respectively. We have added the corresponding discussion to the second paragraph below Table 1 on P.12.

#2: We did not do the reblocking to decorrelate the sample. The samples are obtained after each Monte Carlo step where we defined a single Monte Carlo step in classical models as an update that attempted to flip all sites once, while in Hubbard model as an update that attempted to flip 10% of the auxiliary field. We have added the above description to the last paragraph of Section 2 of the revised manuscript.

#3: In the revised manuscript, we also examined the system size dependence of the Ising model on a square lattice and the result is presented in Appendix A.4. The step size m = 10 is used there and other settings are kept the same as those in Fig. 2. We found that the critical temperature as extracted by fitting the deep learning model output with a tanh function still agrees reasonably well with the theoretical predicted Tc even if the system size goes down to 8x8 sites, where the 20 randomly selected sites account for about 30% of the system size. However, we would like to point out that for smaller system sizes, the machine output fluctuates more around the transition point as compared to large systems. We have included the above discussion in the last paragraph of P. 6.

#4: We thank the referee for the suggestions. In the revised manuscript, we replaced Fig. 3 with the transition temperature extracted by fitting the machine output with the function a.tanh(b. x +c)+d, where a,b,c,d are the fitting parameters. The fitted curves for m =10, 20, 30, 40 are plotted in Fig. 2 for visualisation. The original Fig. 3, where the Tc is extracted by linear fitting the data, is now moved to Appendix B. In both cases, the conclusion that Transformer and Bi-LSTM perform better remains unchanged. The solid lines in Fig. 4 are also replaced by the fitted tanh curves accordingly.

#5: The error bars in Fig. 3 correspond to the standard errors in the predicted Tc from 10 training cycles of the deep-learning models. For each number of MC steps m, we trained the models 10 times, and after each training, the Tc is located by fitting the machine’s output to the tanh function. We have clarified this in the paragraph below Fig.3 in the revised manuscript.

#6: We thank the referee for raising up this question. In the revised Fig. 3, we include the predicted Tc for m=40, 60, 80, 100 extracted from the tanh fit. Despite the conclusion that Bi-LSTM and Transformer predict a Tc close to the theoretical value and they perform better than CNN and FCN remains true, the previous observation, which was drawn from linearly fitting the machine outputs, that the predicted Tc from Bi-LSTM/Transformer converges faster to the theoretical value becomes less obvious. Moreover, from the plot, it seems to suggest that there is an optimal number of steps (e.g. m=10 and m=80 in the case shown) that the machines give better predictions. We have modified the first paragraph below Fig. 3 accordingly to avoid causing any potential confusion.

#7: In Appendix A.2 of the revised manuscript, we have added a more detailed discussion on the effect of the number of elements/sites on the machine’s performance. From Fig. 6, we found that if the number of sites is too small (e.g. 5 sites), the output of the deep learning models has larger fluctuations and they also have less confidence in the prediction even for temperatures far away from the transition temperature, where the output should be either 0 or 1. The situation improves if more elements/sites are used. However, further increasing the number of sites beyond 20 does not result in a significant improvement in the predicted value of Tc. This holds for various system sizes we have considered. Thus, considering the complexity and the implementation of the deep learning models, and the amount of memory usage, we picked 20 sites to form our training samples for the cases discussed in the main text.

#8: We thank the referee for pointing the typo out. We have corrected the typo in the revised manuscript.

#9:
Thank you for pointing out that the issue is also found to happen in other studies. We have followed the referee's suggestions to test whether a longer time series or more elements of Green's function helps to improve.
When we look deeper into the scenario, two factors are contributing to the larger fluctuation in the quantum model. Firstly, unlike the spin configuration in classical models, the input of Green's function does not obey translational symmetry, so the behaviour of the time series sequence depends a lot on the choice of the element’s location. This will easily lead to misjudgement if we pick the element with a very slight difference in the two phases. Secondly, as mentioned, the element in the Green's function fluctuates a lot, especially when close to the phase transition point due to quantum fluctuation. The large fluctuation in input further hinders the deep learning model from classifying the phase correctly.
We find that the referee's suggestion of increasing the input sequence length could help relieve the fluctuation. We further investigate whether increasing the input sequence length from 10 to 100 and 400 or increasing the number of input sequences from 20 to 40 can reduce the fluctuation when the system size is L=6. Figure 10 shows the difference between the output of the deep learning model and the corresponding tanh fit. The larger the difference is, the stronger the output fluctuation of the deep learning model is. In Fig. 10, it can be seen that the closer to the transition point, the larger the difference is. In Fig. 10 (a) and (b), our result shows that increasing the sequence length can reduce the difference mildly. However, we cannot see obvious improvement when we increase the number of input sequences in Fig. 10 (c) and (d). We have added Appendix C to include this further investigation and edited the first paragraph on P.12 to discuss this.

#10: In Appendix A.1, we have added the results for the XY model and the Hubbard model as Fig. 5 (c)-(f) to address the referee’s question. The tanh fit to the machine output (solid curve) estimated a similar Tc in both the unshuffled and shuffled cases, suggesting the statement that the positional embedding does not significantly change the deep learning model’s performance still holds in the XY model and the Hubbard model.

#11: We thank the referee for drawing our attention to the typo in the label. We have corrected it in Fig. 4 (b2) in the revised manuscript.

#12: We have made the raw data and codes publicly available and have included the link (https://github.com/ParcoDing/Rapid-detection) at the end of the revised manuscript.

#13: We thank the referee for pointing this out. We have proofread the manuscript again and corrected the spelling/ grammatical mistakes we found.

---

## Round 2 · Referee Report · Anonymous · 2022-7-17

Report
1- The paper now is well structured and redundancies are removed.
2- The authors address the problem with the linear regression. The extraction of the transition points is now much clearer. The guide to the eye is now replaced with the actual fit. This strengthen the data and gives an idea about the precision of the fit by only looking at the plots.
3- Thank you very much for clarifying this.
4- The explanation and the structure revision of the article now makes everything clear.
5- This is a good comparison and increases the significance of the work.
6- See requested changes. The rest of the article reads much better now.
7- Thank you for sharing this.
I recommend publication and learned a lot from the paper.
Requested changes
1- Last sentence of the second last paragraph on page 3 needs to be reordered or rewritten. No big deal.

---

## Round 2 · Referee Report · Anonymous · 2022-7-22

Report
I thank the authors for the careful discussion and investigation of my questionings. I am happy with the responses and consider the paper to be in good shape. I recommend it for publication.
Below I briefly comment each of the authors replies to the issues I had raised.
#1: Thank you for including Table 1 and appendix D. These add valuable information to the manuscript.
#2: Thank you for clarifying this.
#3: In appendix A.4 the authors wrote that "the extracted Tc is insensitive to the system size and agree well with the theoretically predicted value". I have trouble understanding this result, since I would expect the Tc estimate to be size-dependent. But perhaps such dependence is hidden by the error of the method's Tc estimate. Anyway, I am happy with the discussion of this issue added by the authors. Further investigations of this matter may be addressed in subsequent works.
#4: Thank you for this change.
#5: Perfect.
#6: Thank you for improving this figure. Looking at its new version, it looks like the Bi-LSTM estimate is largely unchanged for 20 or more MC steps with very small error bars. I wonder if this means the Bi-LSTM estimate can be systematically improved with the number of MC steps, even if in this case the converged estimate is slightly biased. Again, a smaller question that might be worth further investigation in future works.
#7: This is valuable information. Thank you for addressing it.
#8: Thank you.
#9: Thank you for further investigating this matter and more carefully discussing it in the text. I am perfectly happy with it. Again, it would be interesting if future works attempt to find ways of circumventing this issue and improve critical point estimates.
#10: Thank you for adding this information and discussion.
#11: Thank you.
#12: Perfect! Thank you.
#13: The text reads much better now. I still came across a couple of minor spelling mistakes that I list bellow.
Requested changes
As I have just mentioned, a few spelling issues remain:
1 - There are a couple of occurrences of the expression "for examples" across the text.
2 - In introduction, P-2, paragraph 3, where the text reads "Our method being discuss here..." should be read "discussed".
3 - At the last paragraph of the introduction, P-3, the sentence "The paper is organised as the followings." should read "The paper is organised as follows.".
4 - First paragraph of P-8 reads "The temperature corresponds to a fitted value...". My interpretation is that it should read "The temperature that corresponds to a fitted value...", otherwise it might hinder the readers understanding.
5 - First paragraph of P-11 reads "the transnational symmetries" while it should read "the translational symmetries".
6 - Second paragraph of P-11 reads "as compared to the method of calculating the order parameters and supervised learning". To me this sentence sounds a bit confusing so I would suggest it be rewritten into something like "as compared to the traditional method of estimating the transition from order parameters and to the supervised learning method".
7 - First paragraph of Conclusion, P-13, reads "before equilibrium and located the critical points" but should read "before equilibrium and locate the critical points".
8 - In Appendix A.4 the text reads "the step size used here is m = 10". This refers to the number of steps taken from the beginning of the MC walk. The way in which it is written might cause confusion on distracted readers (with for example, "time step" of projector quantum MC methods). It could thus perhaps be improved.

---

## Round 2 · List of Changes

1. The detailed description of our method in Sec. 1 is removed and the details are left to Sec. 2 to avoid redundancy.
2. The discussion of the embedding layer in the second paragraph of Sec. 2 is modified.
3. The meaning of one (quantum) MC step is now defined in the last paragraph of Sec. 2.
4. Solid lines in Figure 2 and Fig. 4 are replaced by the tanh fits of the corresponding machine output.
5. Figure 3 is replaced by transition temperatures extracted from tanh fit of the machines’ output, and the corresponding discussion on P.8 is also modified accordingly. The original Fig. 3 where the transition temperatures were extracted from a linear fit is moved to Appendix B.
6. Further discussions on the large fluctuations observed in the vicinity of the critical point in the Hubbard model (Fig. 4(d3)) is added in the second last paragraph of p.12 and Appendix C.
7. Table 1 is added to compare the time cost by using conventional methods and our method in detecting the phase transition points and the corresponding discussion is added to the last paragraph of Sec. 4.
8. Results of shuffling the input series of XY model and Hubbard are added to Appendix A.1.
9. Appendix A.2 and Appendix A.4 is added to discuss the machine performance on varying the number of randomly selected sites and the size effect, respectively.
10. Appendix D is added to include the details of the sample collection for obtaining the data in Table 1.
11. The raw data and codes have been made publicly available at the link https://github.com/ParcoDing/Rapid-detection.

---

## Round 3 · List of Changes

1. Corrected spelling and grammatical mistakes.
  2. Rephrased the last sentence in the first paragraph of Section 2.

---

## Editorial Decision

published